

**Formation of a multi-translational reactivated ancient landslide in the Three Gorges**
**Reservoir, China**
Shilin Luo[1,2], Xiaoguang Jin[1], Da Huang[3*], and Tantan Zhu[1,3]
[1]State Key Laboratory of Coal Mine Disaster Dynamics and Control, Chongqing University,
Chongqing 400044, China
[2]Provincial Key Laboratory of Safe Mining Techniques of Coal Mines, Hunan University of
Science and Technology, Xiangtan, Hunan, 411100, China
[3]School of civil and transportation engineering, Hebei University of Technology, Tianjin 300401,
China
* Corresponding author: dahuang@hebut.edu.cn
**Abstract**
The fluctuation of water levels and seasonal rainfall in a reservoir may induce various types of slope
movements. Some of these movements are new, whereas others are old but reactivated. The primary
aim of this study is to investigate the formation mechanism and process, deposit characteristics, and
identification signs of a giant multi-translational reactivated ancient landslide in the Three Gorges
Reservoir region based on field observations, on-site surveys, and Electron Spin Resonance
experiments. The Outang landslide, located at the south bank of the Yangtze River, has a total
volume of approximately 90 million m$^3$ and can been divided into three independent subzones with
an apparent age of 120–130 ka (ka represents a thousand years) for subzone O1, 65–68 ka for
subzone O2, and 47–49 ka for subzone O3. The features of mobilized material structure and slip
surface morphology in each subzone are similar and are in the form of a spoon. A conceptual model,
including sliding, bending, suspending, and accumulating, is deduced to explain the formation
mechanism and evolutionary process of this instability. Three types of evidences are proposed to
recognize the ancient landslide. Currently, landslide stability is obscure based on the significant
landslide movement and reactivated features; more attention and long-term monitoring is necessary
in the future.
**Keywords:** Three Gorges Reservoir; Ancient landslide; Formation mechanism; Evolutionary
process; Recognition evidences






**Introduction**

The landslide, defined as a wide variety of processes that result in the downward and outward movements of slope-forming materials (Varnes, 1978), is a typical and destructive geo-hazard (Liu et al., 2016; Huang and Zhu 2017;Sättele et al., 2017; Jacobs et al., 2018). Typically, slope instability can be classified into the occurrence of new landslides and the reactivation of ancient landslides, both of which are extremely prevalent in reservoir areas and may create landslide dams, bury residential houses, and pose a significant threat to the natural environment (Gutiérrez et al., 2015; Gu and Huang, 2016; Huang et al., 2018). A well-known reservoir slope instability event is the 1963 Vajont landslide in Italy, where more than 2,000 deaths were reported and several villages were destroyed. Since then, reservoir-induced landside problems have received particular attention from engineers and geologists (Barla, 2013; Mantovani and Vitafinzi, 2003; Wolter et al., 2016).

For flood control and hydropower generation, one of the largest civil engineering projects in human history—the Three Gorges Water Conservation and Hydropower Project in China—was constructed and completed in 2008. Subsequently, the reservoir water level exhibited a cyclical fluctuation of 30 m (145–175 m) under normal operating conditions, and resulted in a 660-km long and 1.2-km wide (on average) hydro-fluctuation belt in the Three Gorges Reservoir (TGR) region extending from Yichang City to the Chongqing Municipality, China (Fig. 1a). Owing to the complex geological environment, climate condition, and reservoir operation, the TGR region has developed into a landslide-prone area (Wang et al., 2016), where more than 5000 reservoir-induced landslides have been identified since the first trial impoundment in 2003 (Jian et al., 2009, Yin et al., 2016, Miao et al., 2014). Studies regarding the trigger mechanisms and factors, deformation characteristics, and motion modes of reservoir-triggered landslides have been published widely (Sun et al., 2016b; Jian et al., 2009; Hu et al., 2015; Chen et al., 2015). Among the reservoir landslides, problems concerning reactivated ancient landslides including reasons for landslide reactivation, risk of a fast sliding, and available emergency civil protection actions have received significant attention, and were investigated once the old instability was identified, e.g., the Quchi landslide (Gu et al., 2017), Huangtupo landslide (Wang et al., 2016), Anlesi landslide (Jian et al., 2009), and Zhujiading landslide (Hu et al., 2015).





However, the clarification of the reservoir ancient landslide formation mechanism, evolution process, and identification evidence are difficult and challenging because such instabilities vary in size, type, disaster-pregnant setting, and formation age (Zhao et al., 2016). In fact, many reservoir-induced old landslides were formed thousands of years ago and have been subjected to long-term reconstructions or sediment covers; further, the original landslide geomorphic features were blurred and modified (Deng et al., 2017; Zhao et al., 2016; Zhang et al., 2018). Such landslides have hardly been identified owing to significant changes in the original features and failure topography. Many researchers have studied the formation and development characteristics based on geological processes (Cruden and Varnes, 1996), the evolution of river valleys (Xu et al., 2008; Yin et al., 2010; Zhao et al., 2007), paleoclimatology (Xie et al., 1992; Yin et al., 2010). However, the understanding and cognition on ancient landsides are rather superficial because of the concealment of ancient landslides, the complexities of their formation mechanism, and the limitations of research methods (Zhang et al., 2018).

The objective of this study is to investigate the formation mechanism and multi-translational process of the Outang ancient landslide through a combined analysis of field investigation, geological exploration, and Electron Spin Resonance (ESR) experiments. We anticipate that this study will provide some important insights into the deposit characteristics of the old instability, as well as the corresponding identification signs.

**Regional geological background**

The Outang landslide (Fig. 1c), a giant reactivated ancient landslide, is located in Anping town of Fengjie County, Chongqing Municipality, China, and is approximately 177 km upstream of the Three Gorges Dam (Fig. 1a). This landslide area belongs to a secondary tectonic unit of the upper Yangtze platform, which is situated on the intersection between the Daba Mountain bow-like folding belt and the east Sichuan folding belt. The strike of geological structures in the study area (marked by rectangle in Fig. 1a and mapped in Fig. 1b) is dominated by the NE–SW direction. The engineering geology map of the Outang landslide is shown in Fig. 2a. The climate in the study area belongs to the monsoon of the subtropical moist climate zone with recognizable four seasons. The average annual air temperature and rainfall are 16.3 °C and 1147.9 mm, respectively. Rainfall is always concentrated in June to September annually and accounts for approximately 70%





of the total annual rainfall in this period.
**Overview of Outang landslide**
**Spatial geomorphological features**
The Outang Landslide, characterized by a reclining bell-shaped surface topography with a
maximum length of 1.8 km and a thickness of about 50.8 m, is located at the south bank of the
Yangtze River and approximately 12 km upstream away from the Fengjie urban area (Fig. 1b). The
instability deposit covers an area of 1.78 km$^2$ with an estimated volume of $9.0 \times 10^7$ m$^3$. It extends
from the front elevation of 90–102 m above sea level (a.s.l.) to the crown elevation of 705 m a.s.l.
along a 75–80° direction from the flow direction of the Yangtze River. The primary slide direction
is about 345°. After a careful observation, a chair-like geometry, characterized by a flat and broad
terrain at the front section but steep terrain at the middle and upper sections, occurred repeatedly
from the water level to the rear part of the landslide in spatial morphology. As illustrated in Fig. 3a,
a slope gradient of 5–10° appears and distributes at the elevation ranging from 160 m a.s.l. to 220
m a.s.l., where Anping town is located and the Fengjie–Anping road runs through; subsequently,
the steep slope, extending to about 320 m a.s.l., occurs with the slope gradient of 20–35°
(occasionally up to 50°). Another flat area (Fig. 3c), located at the elevation of roughly 330 m a.s.l.,
and a cliff (Fig. 3b, the frontal boundary of subzone O2 and mentioned in the chapter 4.2), were
recognized. Similarly, at the altitude of approximately 428–450 m a.s.l., a flat area with a slope
gradient of 2–10° (Fig. 3e) and a cliff (Fig. 3d, the frontal boundary of subzone O3 and mentioned
in the chaptered 4.3) were also identified. The spatial geomorphological features imply that the
Outang landslide might be composed of multiple landslides.
**Lithostratigraphy survey**
The lithostratigraphy of the landslide was studied through geological explorations and field
observations. The mobilized materials can been divided into two layers: shallow colluvium in the
top and fractured sandstone in the deep: 1) The top layer is a mixture of clayey soil and rock
blocks (approximately 25–66% volume content, sandstone, and siltstone) of sizes of 1–40 cm (Fig.
4a). Its thickness (0.2–35 m) is increased gradually toward the toe. 2) Deep in the landslide body,
the primary material is a fractured sandstone layer (Fig. 4b) with varying thicknesses between 10
m and 95 m (occasionally exceeding 110 m at the toe) and is cut intensely by two sets of fissures,



whose orientations follow 120–150°/55–75° (dip/dip angle) and 40–70°/60–85°. Trench and adit
explorations disclosed that dark gray claystone and clayed soil (coal and shale can also been found
occasionally) are predominant at three weak interlayers (WIs) with a thickness of 5–20 cm (Fig.
4c). The bedrock at the attitude of 335–350°/18–24° (dip/dip angle) are constituted by
sublitharenite of the Xujiahe formation in the Upper Triassic system ($T_{3xj}$) and fine sandstones of
the Zhenzhuchong formation in the Lower Jurassic system ($J_{1z}$), with the former overlain by the
latter (Fig. 3a). As shown in Fig. 4d, bedding planes that indwell in the outcrops of the bedrock
(sandstone) at the rear part develop a downslope with the attitude of 336°/18°.
**Weak interlayers**
The presence of a shear plane is often assumed as evidence of a slip surface (Hutchinson and
Bhandari, 1971; Corominas et al., 2005). In-site investigation found three WIs (numbered WI1,
WI2, and WI3, and marked in Fig. 5a). The slip surfaces existing in the WIs were discovered with
clear striated polished surfaces by adit (WI1 in Fig. 5b) and trench explorations (WI3 in Fig. 5c) .
The main mineral materials in the WIs are quartz and clay minerals with the average contents of
more than 35.2% and 44.7%, respectively. Additionally, the clay minerals, composed of
montmorillonite (up to 74%), illite (15%–31%), and kaolinite (5%–11%) are characterized by high
swelling, softening potential, and low permeability. To clarify the apparent age of the Outang
landslide, the ESR experiment was conducted and eight samples near the slip surface were
prepared using a drilling hole, trench, and adit explorations (marked in Fig. 2a). Information on the
eight samples is presented in Table 1. The ESR results indicate (in Fig. 2a) that the apparent age of
the Outang landslide can been subdivided into 120–130 ka for the low part, 65–68 ka for the
middle part, and 47–49 ka for the upper part, thus proving that the ancient instability could be
composed of several independent landslides.
**Activity signs**
Since the first trial impoundment, the instability activity features have became increasingly
obvious. At the low part, most of its volume is submerged by the reservoir water owing to its
low-flat terrain and local collapse, and failures were observed in the reservoir water fluctuation
zone frequently. Figure 6b illustrates the representative local collapse with a material of volume of
$2.3 \times 10^4$ m$^3$ sliding into the river, and the front boundary of the landslide retreated nearly 6 m with





some falling sandstone blocks observed on the slope surface. The movement has also resulted in
some damages in the Anping town houses (Fig. 6c). At the middle part, a long tension crack of
length 74.5 m, width 0.1–75 cm, and visible depth 10–110 cm was observed at an altitude between
350 m and 370 m a.s.l. (Fig. 6d). Meanwhile, the dislocation and cracking of the road at a few
places were identified. For example, Fig. 6e shows a typical dislocation with the maximum
dislocation of 0.5 m. At the upper part, many signs of reactivation were also exhibited. A fallen
telegraph pole was found as a consequence of the continuous surficial movement (Fig. 6f); the
newly installed telegraph pole was inclined downslope at an angle of 12° (Fig. 6f). The upslope
boundary of the instability is defined by a scarp, where WI3 was exposed (Fig. 6g).
**Accumulation characteristics of active parts**
In terms of the spatial geomorphology features, ESR results, and slip surface morphology, the
Outang ancient landslide can be divided into three independent reactivated subzones (labeled
subzones O1, O2, and O3). Generally, the component of the mobilized material in each subzone is
similar with that of the top layer is shallow colluvium material (clayey soil and rock blocks), and
the deep layer is fractured sandstone. The fractured sandstone structure is spoon like, characterized
by an orientation of stratified or stratoid bedding planes (335–350°/18–24°) from the rear to
mid-fore part; it changes to nearly horizontal, and curves upward at the toe area (155–170°/0–15°)
in each subzone. For example, the orientation of the bedding planes within the fractured sandstone
is 340°/21° (Fig. 7b), but changes to 162°/15° (Fig. 4b). Moreover, the variation rule of the
fractured sandstone structure in each subzone is similar to its respective slip surface morphology.
**Subzone O1**
Located at the low part of the landslide, subzone O1 has a reclining bell-shaped surface
geometry with a primary slip direction of roughly 345° (Fig. 2a). The elevation of the frontal part of
subzone O1 is approximately 90–102 m a.s.l. (submerged by water completely); the crown
elevation is 300–370 m a.s.l. and is covered partly by subzone O2. It has a maximum length of 880
m, a width of 1100 m, and an average thickness of 70.3 m. This zone has an area of $9.22 \times 10^5$ m$^2$
and a volume of $6.48 \times 10^7$ m$^3$. As mapped in Fig. 7a, the shallow colluvium material has an uneven
thickness of 10 to 35 m; the layer of fractured sandstone exhibits an average thickness of 62 m. Two
local strong deformation areas distributed at both sides of the toe of O1 (mapped in Fig. 2a) were



clarified with a total volume of $4.1 \times 10^6$ m$^3$. Slip surfaces with clear sliding traces for the two local
strong deformation areas were also revealed by geological exploration (Fig. 7c–d). Moreover, adit
exploration revealed that the slip surface of this subzone existed in WI1, where the main
components were claystone (size of 0.5–6 cm) and clayed soil (content of 60–80%), whose colors
are dark gray and gray, respectively (Fig. 4c).

**Subzone O2**

Subzone O2 is located at the middle of the landslide. It extends from 250–300 m at its toe to
400–530 m a.s.l. at its rear section, where it is wrapped partly by subzone O3. This zone has a
length of approximately 440 m, width of 650 m, area of $3.16 \times 10^5$ m$^2$, and volume of $1.02 \times 10^4$ m$^3$.
As illustrated in Fig. 8a, the thickness of the deposit material in this subzone is approximately 32
m, and the thickness of the shallow colluvium is less than 6 m. The slip surface of subzone O2 also
exists in WI1; it was revealed by a drill hole (numbered 5 in Table 1) of thickness 3–15 cm and
was dark gray. The field investigation shows a cliff with a vertical dislocation of 8.5 m, and the
exposed fractured sandstone is the frontal boundary of this subzone (bounded in Fig. 3a and
illustrated in Fig. 3b); the lateral boundary is a ridge in the west (Fig. 1c and Fig. 3c) and a gully in
the east (Fig. 1c). Owing to the blockage of the rear part of subzone O1, the flat area appears at the
mid-fore part of subzone O2 (marked in Fig. 3a and displayed in Fig. 3c). The ESR experiment
shows that the apparent age of subzones O1 and O2 are 120–130 ka and 65–68 ka, respectively,
indicating that both subzones occurred at different times; specifically, subzone O2 occurred later
than subzone O1.

**Subzone O3**

Subzone O3 was seated at the upper part of the Outang landslide with a length of 0.64 km,
width of 0.83 km, and an average thickness of 27.2 m that increased downslope. It extends from
400–530 m a.s.l. to 705 m a.s.l., with an entire planar area of 0.54 km$^2$ and a volume of $1.45 \times 10^7$
m$^3$. The shallow colluvium is extremely thin (less than 1.2 m, as shown in Fig. 7a). Cliff
daylighting with broken rock mass and bloating with many shear and tension-shear cracks was
found at an altitude of roughly 408 m a.s.l., which is the frontal boundary of this subzone (labeled
in Fig. 3a and illustrated in Fig. 3d). The flat area located at the mid-fore part of subzone O3 was
also recognized (marked in Fig. 3a and shown in Fig. 3e). Trench exploration disclosed that the





slip surface of this subzone existed in WI3, where the dominated component was carbonaceous
claystone (size of 2–18 mm and content of 75%) and dark-gray clayed soil of thickness 8–20 cm
(Fig. 4c). Moreover, the rear part of this subzone is bounded by the scarp of the daylighting of WI3
(Fig. 6g). The ESR experiment indicates that the apparent age of subzone O3 is the latest
compared to that of subzones O2 and O1.
**Discussion**
**Mechanism and process**
Based on the investigation and analysis of the basic characteristics of the Outang landslide, and
in conjunction with the information about the engineering geology and ESR experiments, a
conceptual model for understanding the formation mechanism and multistage sliding process of
this instability is deduced and shown in Fig. 8. The detailed descriptions of these processes are as
follows:
1. At the stage of Early Pleistocene, the whole TGR has suffered from intermittent uplift, tilt,

and continuous action of intensive river incision (the fluvial incision rate up to 92.5 cm/ka

between the river section of Chongqing and Fengjie) as he effect of the Himalayas'

movement, causing the appearance of steep-sided valleys that provided space for the slope

movement (Fig. 8a–b).

2. A weak interlayer (WI1), and some tectonic and weathering cracks within the rock mass

occurred. During heavy rainfall, rainwater could penetrate the rock mass along the cracks

and converge to the WI1. On one hand, the WI1 would suffer from the action of softening

or argillation, and then slip or squeeze out to the free surface (Zhao et al., 2012). On the

other hand, hydrostatic pressure caused by water flow could induce a larger uplift pressure

(Zhao et al., 2015). Both factors contribute to the overlying rock stratum creep–slide along

WI1 and result in tensile cracks (Fig. 8b). Because WI1 is not exposed at the lowest part of

the slope, where the sliding resistance mass is strong, stress will be concentrated at the toe

area of the slope because of the slope creep deformation, thus resulting in a slight bending

and upward uplift of the rock stratum at this part (bounded by rectangle in Fig. 8b).

3. Under the combined action derived from long-term precipitation infiltration and gravitation,

the creep deformation of the rock stratum above WI1 is successive, thus widening the





tensile crack and extending it toward WI1 (Fig. 8c). Additionally, the fracture and flexure
upward of the rock mass become increasingly acute at the toe, accompanied by numerous
fissures and relaxed rock mass (bounded by rectangle in Fig. 8c). The orientation of the
fractured sandstone next to the toe area (Fig. 4b) of subzone O1 is the best evidence to
justify the processes of bend and fracture of the rock stratum. These processes further
cause rainwater to seep into the slope easily and generate a potential curved shear surface
with a tendency to join with the WI1 gradually (Fig. 8c).
4.  As time progressed and owing to the infiltration of rainwater, the tensile structural plane

and potential curved shear surface became gradually connected with WI1. Approximately

120–130 ka ago, as induced by heavy rain and other factors, a translational landslide

(subzone O1) occurred along the shear rupture surface (slip surface) (Fig. 8d). Thus, the

formation mechanism of subzone O1 can be summarized as slide–bending. During the

sliding process, under the influence of the reservoir buttress effect (circled in Fig. 8d)

(Paronuzzi et al., 2013) and slip surface morphology (Fletcher et al., 2002; Sun et al.,

2016), the moved subzone O1 stopped progressively.

5.  After the generation of subzone O1, a new free surface (Fig. 8d) occurred at the front of the

rock stratum at the middle part of the slope consequent, where WI1 is exposed. The

mechanical property of WI1 would further deteriorate under the adverse effect of water;

moreover, the rock stratum at the middle part exhibits a large tendency to move along WI1

on account of the long-term gravitational deformation (Deng et al., 2017). Approximately

65–68 ka ago, another translational landslide (subzone O2) was induced along WI1 (Fig.

8e). Therefore, the formation mechanism of subzone O2 can be summarized as a planar

slide.

Owing to the blockage of subzone O1, the moving subzone O2 gradually stopped with

the rear area of subzone O1 being covered by subzone O2. In this process, part of the
moving rock mass at the front part of subzone O2 is crushed, and accumulates along the
slope surface of subzone O1, whereas most parts of the rock stratum remain on the slope
and/or hang in air (circled in Fig. 8e). Moreover, controlled by the slip surface morphology
and hindrance of the rear part of subzone O1, the dip of the rock stratum at the front part



of subzone O2 is typically opposite to that of the slope, and the flat topographical area can
been recognized (Fig. 3c).
6. The phenomena of unloading, rebound, and attenuated sliding resistance mass at the toe of
the upper part of the slope are inevitable after the emergence of subzone O2; they are in
favor of the creep deformation of the upper part of the slope above WI3 and give rise to
the bending and upward uplift of the rock stratum at the toe of this part (bounded in Fig.
8e). When the possible rupture surface was created and connected with WI3 gradually, as
induced by heavy rain and other factors, a landslide (subzone O3) occurred (47–49 ka ago
(Fig. 8f)). The formation process of subzone O3 is similar to that of subzone O1; the
formation mechanism can be classified as slide–bending. Moreover, for the reason of the
blockage of subzone O2, the moving subzone O3 gradually stopped with the rear area of
subzone 2 covered by subzone O3. The structure of the rock stratum and its morphological
features are also similar to those of subzone O2, as well as a flat topography area can also
be found (Fig. 3e).
For subzone O1, after the landslide, a relatively stable period characterized by a thicker top
layer formed, and the morphology and location of the frontal boundary changed due to the humid
and rainy climate (Fig. 2a and/or Fig. 6a). For subzones O2 and O3, the rock stratum that is pushed
out is suspended without support at its bottom (Figs. 8e–f). Owing to its own gravity and the
weather, local cracking and the collapse failure of rock mass occurred; this may explain the
occurrences of cliffs with large dip angles and many fissures at the front parts of subzones O2 and
O3 (labeled in Fig. 2a and shown in Fig. 2b and Fig. 2b, respectively). Meanwhile, the failed mass
would move downslope and accumulate on the slope surface; this may have caused the thickness of
the top layer for each subzone to follow the order of O1＞O2＞O3. Another reason may be the
apparent age; this implies that the earlier the subzone occurs, the thicker is the top layer. For the
Outang landside, the earliest is subzone O1, followed by subzone O2 and then the latest is the
subzone O1. Overall, with the evolution of the Yangtze River, the long-term geologic force has
evidently changed the features of the original slope. Hence, the Outang landslide is an ancient
landslide that has experienced the long process of sliding, bending, accumulating, and remolding.
**Identification evidence**


Since the occurrence of the old landslide, it has undergone a long-term geological transformation (human activities, weathering, erosion, etc.). Consequently, many original landslide characteristics have vanished, and the original topography and geomorphology have varied, thus causing significant difficulties in landslide identification (Zhao et al., 2015). However, in the specific case of the Outang ancient landslide, some evidences remain that can be used to identify such an old instability.

1. Mobilized material structure characteristics and slip surface

The landslide is located at the southeast wing of the Guling syncline, where the orientation of the bedrock is 335–350°/18–24°. The dip in the bedding planes within the mobilized materials is the same as that of the bedrock from the rear to the mid-fore (335–350° in Fig. 7b), but opposite to that of the bedrock at the toe of each subzone (155–170° in Fig. 4b). This implies that the attitude of the mobilized material is variable, as characterized by the dip angle decreased nearly horizontal and curved upward at the toe of each subzone, which is analogous to its respective slip surface. The slip surface, particularly of the striated polished surface, interpreted as a result of relative displacements among the displaced materials and the bedrock, was revealed by geological survey (Fig. 5b–c) with the orientation similar to that of the mobilized material. Thus, the material structure characteristic and rupture surface are strong and clear evidences for identifying the ancient landslide.

2. Landform characteristics

As previously mentioned, the Outang landslide has experienced the long process of remodeling with a steep topography (varies from 20° to 45°) under the elevation of 160 m a.s.l.; however, it changes to 5–10° immediately at the altitude of 160–220 m a.s.l. with the occurrence of a flat and broad area (a slope of 5–10°), where human activities (e.g., building roads (Fengjie–Anping road) and reclamation projects (Anping Town)) were frequent and intensive. Analogously, two cliffs at the elevation of about 290 m a.s.l. and 462 m a.s.l. (Fig. 3b and Fig. 3d, respectively), and the flat terraces at the mid-fore parts of O2 and O3 (Fig. 3c and Fig. 3e, respectively) have attracted a substantial amount of attention, thus rendering the landslide area significantly different from the surrounding mountains and easily recognized. Thus, topography saltation occurring in the landslide area will be the important evidence of landslide identification in field investigations.





3.  Underground water characteristics

For the Outang landslide, many cracks distributed at the ground surface provided better access

for rainfall infiltration. As shown in Fig. 7a, the thickness of mobilized materials decreased
substantially (less than 1.2 m for shallow materials and approximately 26 m for the fractured
sandstone layer at sliding mass O3). Moreover, the permeability for the fractured sandstone is large
(roughly $3.35 \times 10^{-3}$ cm/s). These factors allow rainwater to sweep easily into the rock mass, thus
causing increasing underground water level that appear in the form of a spring at the front part of
each subzone. Typically, this type of spring is characterized by a large flow during the heavy
rainfall season (Fig. 9a) that decreases substantially during the dry season (Fig. 9b).
**Evolution of stability**

The earliest geological survey report, provided by the Sichuan Geology and Mineral Bureau in

August 1988, indicated that the Outang landslide is stable or quasi-stable, which was further
confirmed by the Comprehensive Survey Bureau of the Yangtze Water Resources Committee in
December 1995. However, the deformation and failure of old landslide have been discovered
frequently and have received particular attention by local residents and authorities; therefore, a
landslide disaster prevention project (installing anti-slide piles, etc. labeled in Fig. 2a and mapped
in Fig. 2b) was proposed by the Yangtze Institute of Survey, Planning, Design, and Research, and
was completed in November 2003. Since then, landslide stability has been improved significantly.
Unexpectedly, after the TGR dam was completed in 2008, the landslide reactivated signs,
including the damage of houses and roads, broadening of cracks, failure of local collapse, etc.,
were increasingly evident. Meanwhile, striated polished surfaces were also discovered by
geological exploration, and two local strong deformation areas with a total volume of $4.1 \times 10^{6}$ m$^3$
distributed at both sides of the toe of O1 was recognized as well. Thus, the landslide stability
decreased substantially and exhibited the tendency of a complete failure, as reported in November
2012. Although another remediation project, including backfill toe weight and lattice revetment in
the east strong-deformation area (bounded in Fig. 2a and shown in Fig. 2c) and masonry revetment
in the west strong-deformation area (marked in Fig. 2a and demonstrated in Fig. 2d), was
completed in 2013, the landslide is in a state of continuous creep deformation with increasingly
evident activity signs hitherto (including road and house damages, ground fissures, local collapses,



etc.). Further, the landslide stability is ambiguous (Huang and Luo 2018, under review).
**Conclusion**
The Outang landslide could be divided into three subzones with an apparent age of 120–130 ka
for subzone O1, 65–68 ka for subzone O2, and 47–49 ka for subzone O3, among which the change
rules of the deposit material attitude in each subzone were similar and the same to its respective slip
surface morphologies. Additionally, two local strong deformation areas were identified at both sides
of subzone O1.
This landslide deposit has evolved from multiple ancient translational sliding masses with the
formation mechanism of slide–bending for subzones O1 and O3, and planar sliding for subzone O2.
Moreover, the local collapse and accumulation downslope, and the apparent age of each subzone
could be the reasons for the change in the thickness of the top layer.
The material structure characteristics, rupture surface, topography saltation, and seasonal
variation of groundwater exposure could be regarded as valid proofs in identifying ancient
landslides during an on-site investigation.
Currently, although the landslide has undergone two remedial measures, its stability remains
uncertain based on the significant landslide deformation and reactivated features. Therefore,
long-term monitoring and emergency civil protection actions are necessary.
**Acknowledgments**
This work is supported by the Fundamental Research Funds for the Central Universities (No.
106112017CDJXSYY002), the Graduate Research and Innovation Foundation of Chongqing,
China (No. CYB17043), the National Natural Science Foundation of China (Nos. 41672300,
41472245 and 51578091), and Open Research Fund Program of Hunan Province Key Laboratory
of Safe Mining Techniques of Coal Mines (No. E21831).

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



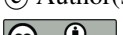

Table 1 The information and images of eight samples

| Number | Picture | Description | Sampling depth/m | Elevation/m a.s.l. | Method |
|---|---|---|---|---|---|
| 1 | | Dark gray clay soil | 183–183.5 | 270.1 | Adit excavation |
| 2 | | Dark gray carbonaceous claystone interbedded with coal | 122–122.9 | 179.8 | Drill hole |
| 3 | | Gray argillaceous clay soil and gravel | 44.5–49.2 | 299.3 | |
| 4 | | Gray claystone interbedded with shale | 41.4–42.7 | 324.4 | |
| 5 | | Dark gray, gray carbonaceous claystone | 32.9–33.6 | 415.2 | |
| 6 | | Gray argillaceous claystone | 29.6–29.8 | 470.1 | |
| 7 | | Gray argillaceous claystone | 40.8–41.1 | 542.9 | |
| 8 | | gray soil | 3.2–3.5 | 446.2 | Trench exploration |



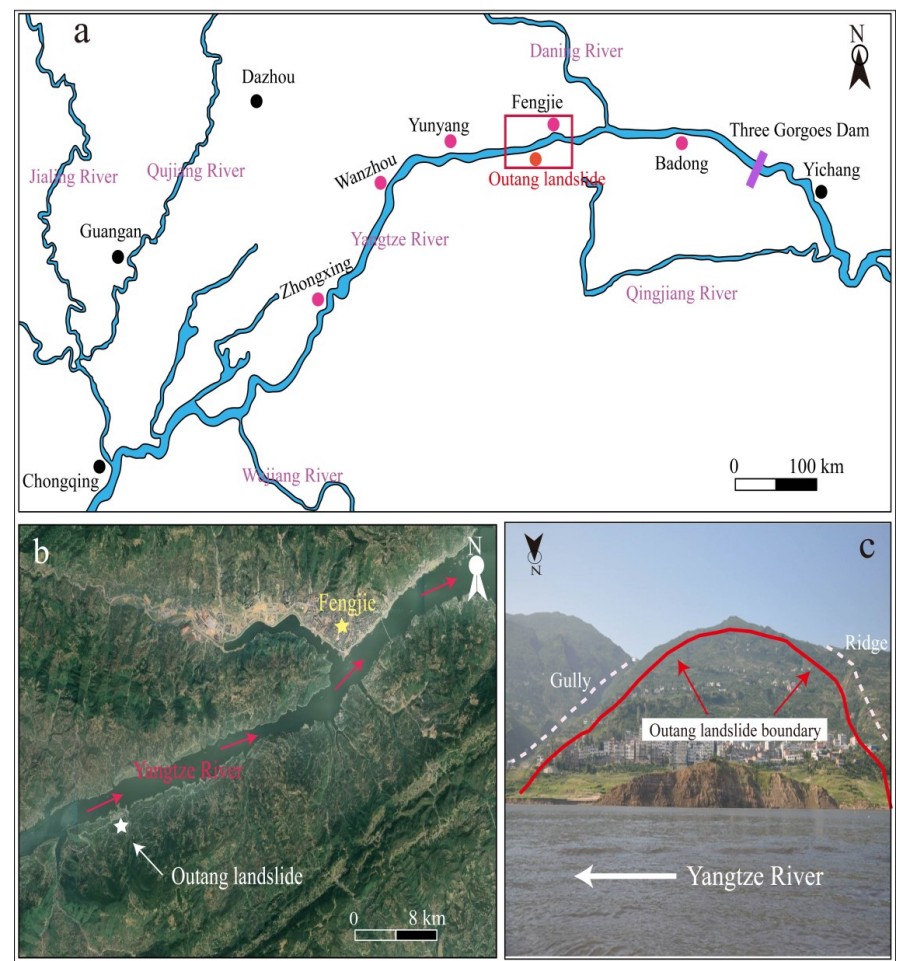

Fig. 1. Outang landslide in the Three Gorges Reservior area. **a** The Three Gorges Reservoir area. **b** Landslide location map of the study area. **c** A closed-up view of the Outang landslide



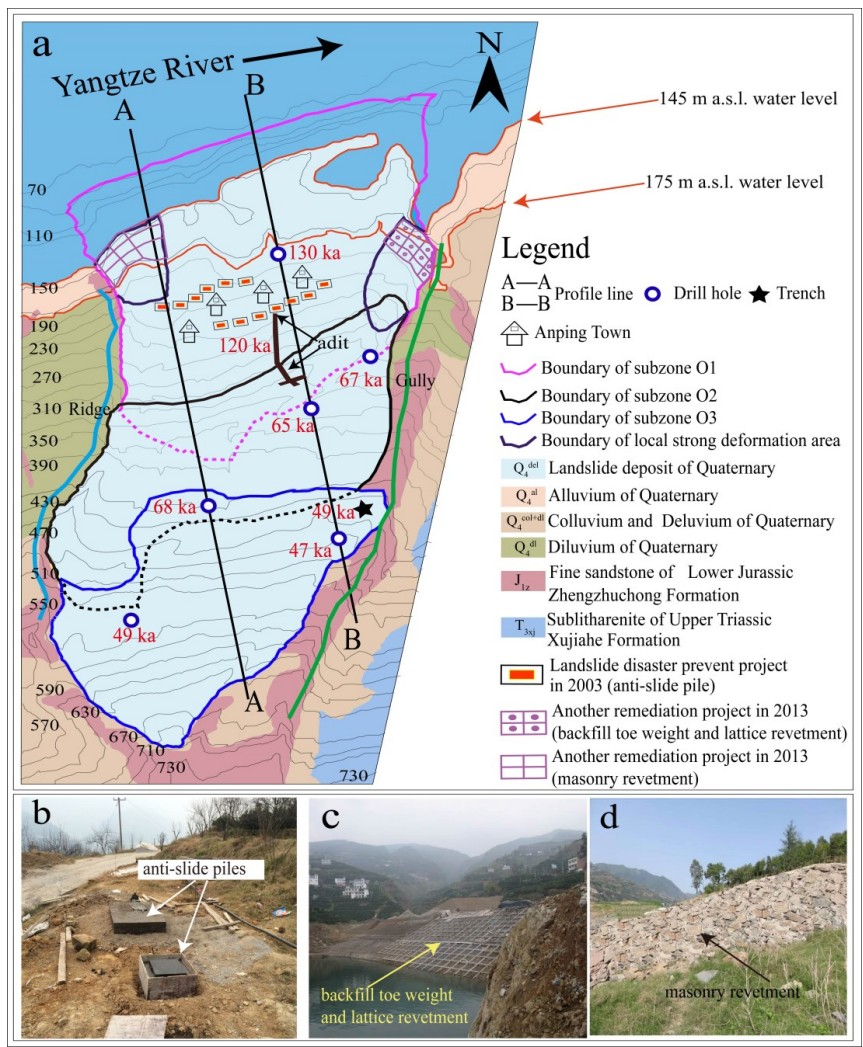

Fig. 2. Outang landslide and some pictures of landslide treatment in 2003 and 2013. **a** Engineering geological map of the landslide. **b** Partial images of the landslide disaster prevent project in 2003. **c–d** Partial photos of the landslide remediation project in 2013


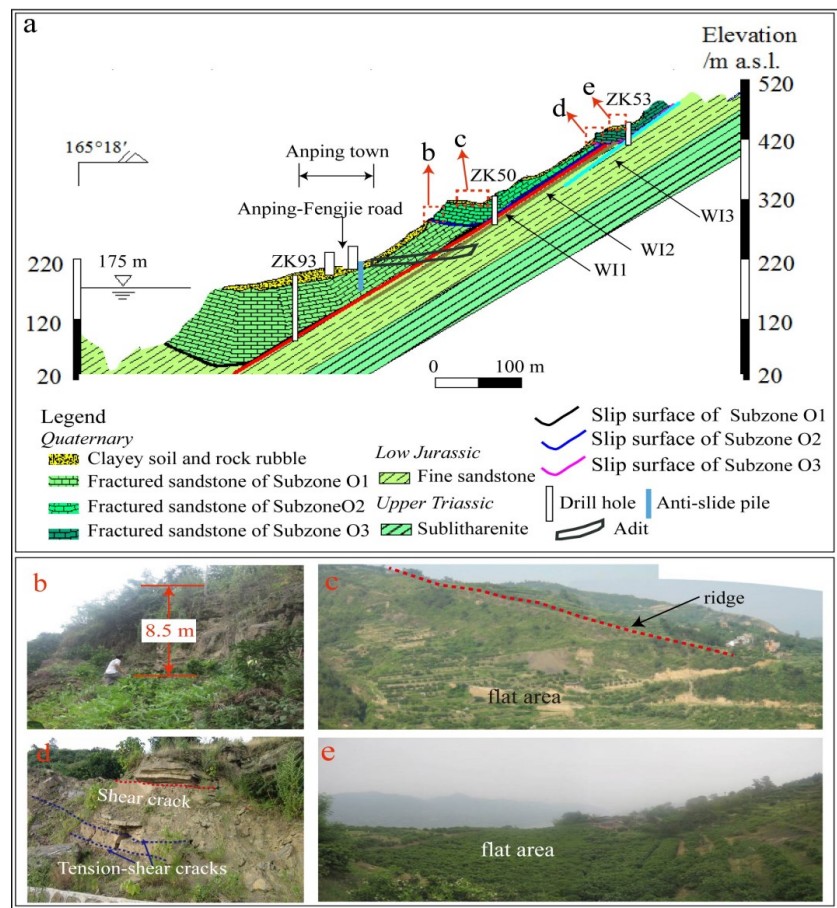

Fig. 3. Outang landslide and its geomorphological features. **a** Geological cross section B-B of the landslide (see location in Fig. 2a). **b** and **d** Partial photographs of cliffs (see location in Fig. 3a). **c** and **e** Images of flat areas (see location in Fig. 3a)




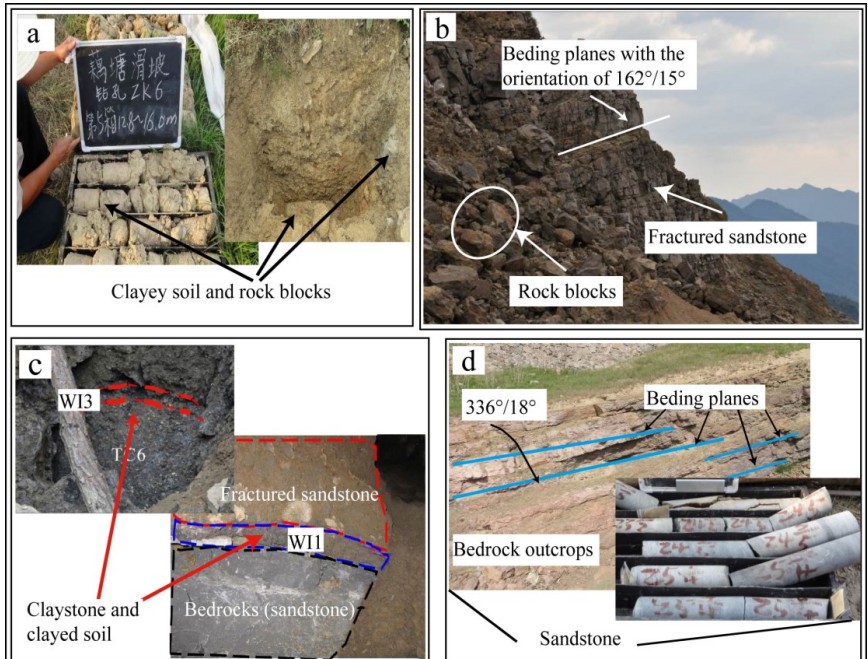

Fig. 4. Bedrock and mobilized materials of the landslide. **a** Shallow colluvium. **b** Bedding planes and rock daylighting at the of the landslide. **c** Weak interlayers revealed by adit (WI1) and trench (WI3) exploration. **d** Bedrock exposed by drill hole and bedding planes within the bedrock daylighting at the rear of the landslide.



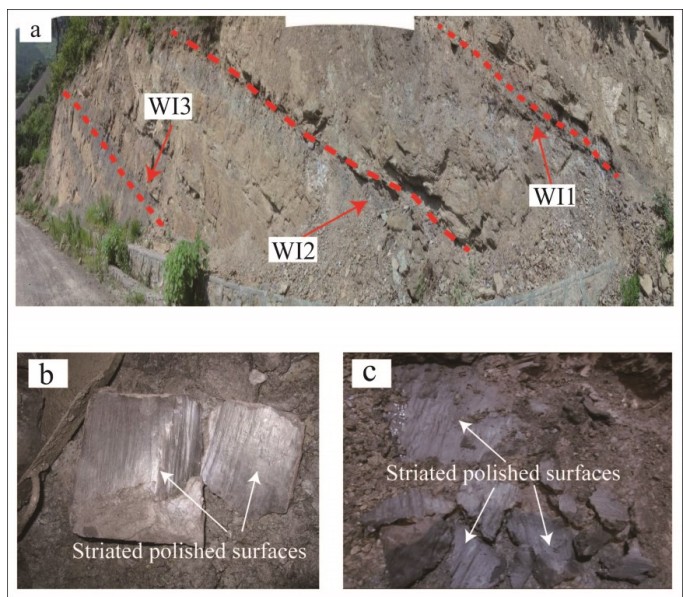

Fig. 5. Weak interlayers. **a** Weak interlayers outcrops. **b–c** Slip surfaces with striated polished surfaces revealed by adit and trench exploration



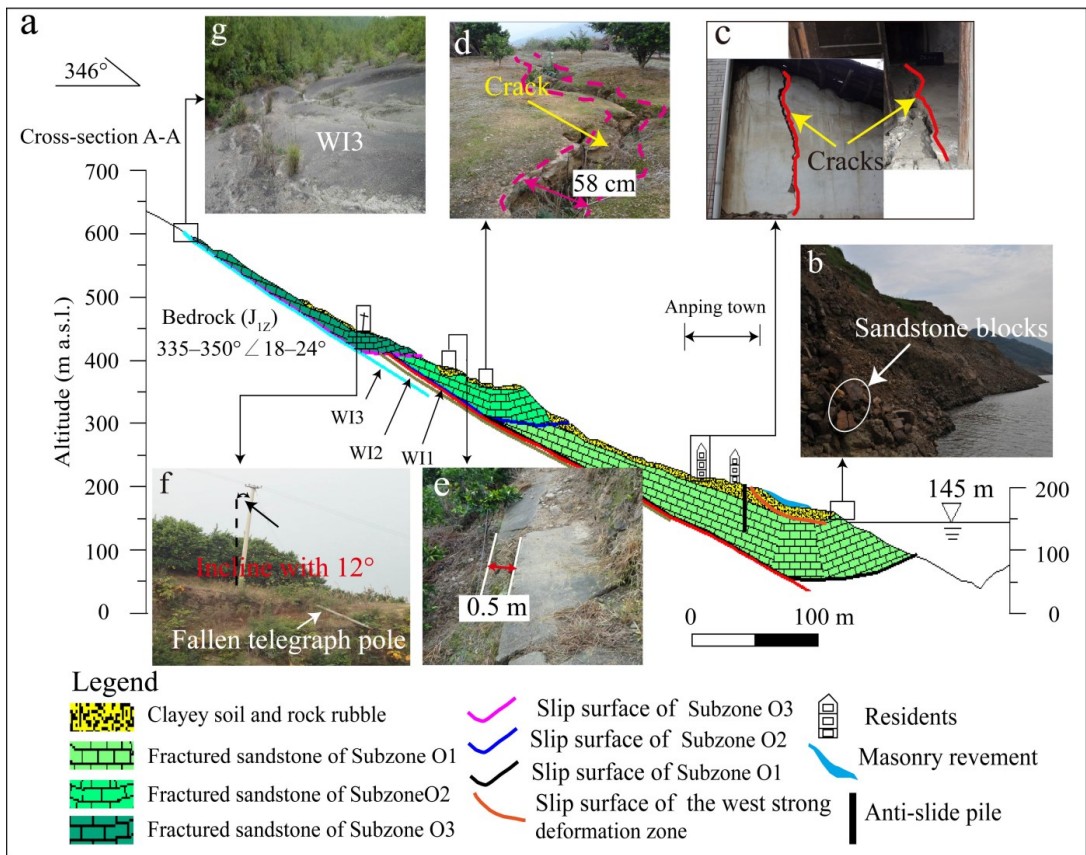

Fig. 6. Geological profile and Failure features on the surface of the Outang landslide. **a** Geological cross section A-A of the instability (see location in Fig. 2a). b Local collapse at the front part of the landslide. **c** Cracks on houses. **d** Cracks on ground surface. **e** Road dislocation. **f** Fallen and inclined telegraph pole. **g** Scrap with the exposure of WI3 at the rear part of the instability



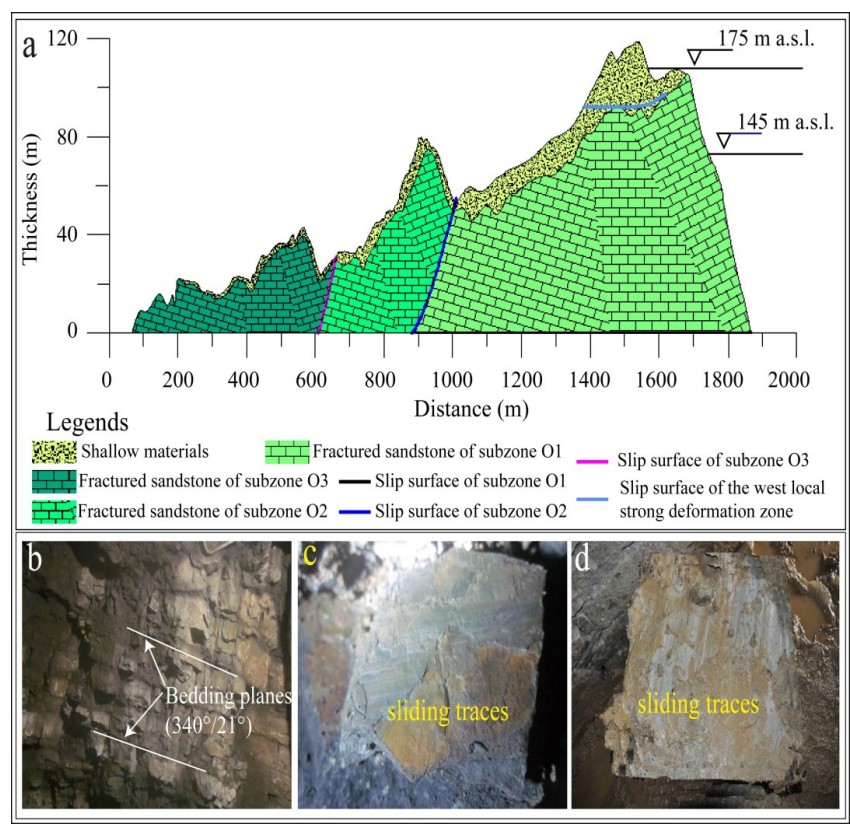

Fig. 7. Outang landslide. **a** The thickness of main body varies with the horizontal distance derived from Fig. 6a. **b** The orientation of the bedding planes within the fractured sandstone exposed by adit exploration. **c–d** Slip surfaces with clear sliding trace for two local strong deformation areas





Fig. 8. The conceptual model of the formation mechanism and process for Outang landslide. **a** Fluvial incision. **b** The phenomenon of the creation of tensile crack and slight bending and upward uplift of rock stratum. **c** The fracture and flexure upward of rock mass and the occurrence of the potential curved shear surface. **d** The formation of the subzone O1 and after that the emergence of the free surface. **e** The formation of the subzone O2 and after that the possible rupture surface with bending and upward uplift of rock stratum. **f** The occurrence of the subzone O3





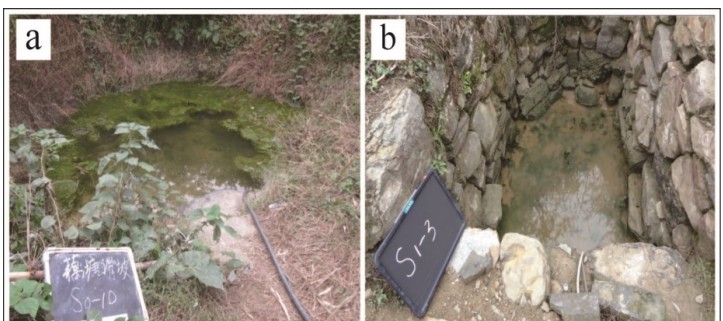

Fig. 9. Seasonal variation of groundwater exposure. **a** Spring at wet season. **b** Spring at dry season