# Peer review of "Formation of a multi-translational reactivated ancient landslide in the Three Gorges"

_Natural Hazards and Earth System Sciences, 2018_

## Referee Comment (RC1) · Anonymous Referee #1 · 9 Feb 2019

The manuscript could be very interesting, but it needs a complete reorganization of the text and a deep check of the English. Section of Methodology and Results are missing. The description of the AoI have to be also reorganized. For this reason, I suggest to reject the manuscript.
* * *

---

## Referee Comment (RC2) · Anonymous Referee #2 · 12 Feb 2019

SHORT SUMMARY The manuscript aims to investigate the development of a reactivated ancient landslide located in the Three Gorges Reservoir (China). The authors analyse and describe the morphological and geological signatures of the landslide in the attempt to reconstruct the failure model and the kinematic history of the landslide.

GENERAL COMMENTS The manuscript deals with an interesting and relevant topic, but the manuscript has to be completely reorganized. The text seems a technical report more than a scientific paper. The description of the study area need a new organization and could be shortened. I would suggest to include also a section Methodology and a section for the Results. Reading the text was challenging and the fragmentation in

many sections and subsections decreases the scientific value of the topics covered. For this reason, I suggest to reject the manuscript. In the following some specific comments are indicated, the authors words are in quotation marks (""). The sections are indicated wiht number and subsections with letters.

SPECIFIC COMMENTS 1. Introduction "The landslide, defined as a wide variety of processes that result in the downward and outward movements of slope-forming materials." This definition was extracted from https://pubs.usgs.gov/fs/2004/3072/pdf/fs2004-3072.pdf, not from Varnes, 1978. "Typically, slope instability can be classified into the occurrence of new landslides and the reactivation of ancient landslides." Not exactly. Please verify in Varnes, 1976, Cruden and Varnes 1978 and Hunghr et al. 2014. "Well-known reservoir slope instability event is the 1963 Vajont landslide in Italy, where more than 2,000 deaths were reported and several villages were destroyed. Since then, reservoir-induced landside problems have received particular attention from engineers and geologists (Barla, 2013; Mantovani and Vitafinzi, 2003; Wolter et al., 2016)" I would suggest to read the last paper Dykes and Brombhead, 2018.

2. Regional geological background The description of the background includes also other information not only the geological one. "The engineering geology map of the Outang landslide is shown in Fig. 2a." It is not necessary here. For the landslide there is a specific section.

3. Overview of Outang landslide This section results in a fragmented description of the landslides (the five subsections listed in the following). I would suggest the authors to reduce and try to organized this section focusing only to the data useful to explain the methodology and to the discuss the results. A. Spatial geomorphological features "The Outang Landslide, characterized by a reclining bell-shaped surface topography." The sentence is not clear. B. Lithostratigraphy survey C. Weak interlayers D. Activity signs E. Accumulation characteristics of active parts

5. Discussion The lack of a methodology doesn't allow understand which are the results. Again the discussion is not a discussion but a very confusing presentation of knowledge (the three subsections are listed on the following), some results and also the discussion about the state of activity of the landslide (in the section "evolution of stability") is a sort of summary of the results but not a real discussion. In particular, the discussion about the state of the activity of the landslide should be the main topic but the authors simply state "the landslide stability is ambiguous". This is not acceptable since the authors have dealt with many data and information. F. Mechanism and process G. Identification evidence H. Evolution of stability

6. Conclusion The conclusion is a synthesis of the paper and in closing the authors highlight the need of monitoring measurements due the uncertainty of the landslide deformation and reactivation without suggesting which type of monitoring techniques could be suitable for such type of landslide.

Figure 1 1a: Please add coordinates or an index map. 1a: Please use the same symbols of figure 2b to indicate Fengji and Outang landslide. Figure 2 2a: Please add a scale bar. 2a: Please emphasize the labels inside the patches of the legend. Figure 3 3a: The line O3 in the profile is not visible.

---

## Editor Comment (EC1) · Elena Petrova (Editor) · 16 Feb 2019

The manuscript investigates an interesting and relevant topic: the development of a reactivated ancient landslide located in the Three Gorges Reservoir (China). The manuscript gives a geological background of the region and a very detailed overview of Outang Landslide. However, sections of Methodology and Results, which are essential for scientific papers, are missing. Discussion section is not a real discussion. The manuscript needs a thorough check of the English. Both reviewers suggest to reject the manuscript.

The editor agrees with the reviewers and suggests to reject the manuscript in this ver-

sion. The authors can completely revise the manuscript according to the reviewer comments and resubmit it to the NHESS again. The text has to be restructured: sections of Methodology and Results have to be included in the manuscript. The description of the study area have to be shortened and reorganized. The Discussion section should really discuss the Results obtained by the authors. The English of the manuscript should be copyedited.